

# Data-driven methods to estimate the committor function in conceptual ocean models

Valérian Jacques-Dumas[1,2], René M. van Westen[1], Freddy Bouchet[3], and Henk A. Dijkstra[1,2]

[1]Institute for Marine and Atmospheric research Utrecht, Department of Physics, Utrecht University, Utrecht, the Netherlands
[2]Centre for Complex Systems Studies, Department of Physics, Utrecht University, Utrecht, the Netherlands
[3]Univ. Lyon, ENS de Lyon, Univ. Claude Bernard, CNRS, Laboratoire de Physique, Lyon, France

**Correspondence:** Valérian Jacques-Dumas (v.s.jacques-dumas@uu.nl)

**Abstract.** In recent years, several climate subsystems have been identified that may undergo a relatively rapid transition compared to the changes in their forcing. Such transitions are rare events in general and simulating long-enough trajectories in order to gather sufficient data to determine transition statistics would be too expensive. Conversely, rare-events algorithms like TAMS (Trajectory-Adaptive Multilevel Sampling) encourage the transition while keeping track of the model statistics. How-
ever, this algorithm relies on a score function whose choice is crucial to ensure its efficiency. The optimal score function, called committor function, is in practice very difficult to compute. In this paper, we compare different data-based methods (Analogue Markov Chains, Neural Networks, Reservoir Computing, Dynamical Galerkin Approximation) to estimate the committor from trajectory data. We apply these methods on two models of the Atlantic Ocean circulation featuring very different dynamical behavior. We compare these methods in terms of two measures, evaluating how close the estimate is from the true committor,
and in terms of the computational time. We find that all methods are able to extract information from the data in order to provide a good estimate of the committor. Analogue Markov Chains provide a very reliable estimate of the true committor in simple models but prove not so robust when applied to systems with a more complex phase space. Neural network methods clearly stand out by their relatively low testing time, and their training time scales more favorably with the complexity of the model than the other methods. In particular, feedforward neural networks consistently achieve the best performance when trained with
enough data, making this method promising for committor estimation in sophisticated climate models.

## 1 Introduction

Global warming may lead to the destabilization of certain subsystems of the climate system, called tipping elements (e.g. Lenton et al., 2008). A recent inventory of these tipping elements, including the Amazon rainforest and the polar ice sheets, has provided estimates of the their critical temperature thresholds (Armstrong McKay et al., 2022). Two important tipping
elements involving the ocean are the subpolar gyre convection and the Atlantic Meridional Overturning Circulation (AMOC).

In the case of the AMOC, the salt-advection feedback is known to cause a bistable regime in conceptual ocean models (Stommel, 1961; Rahmstorf, 1996). Many model studies have shown that the AMOC may collapse under a changing freshwater forcing, either by crossing a tipping point, or by noise or rate-induced transitions. Observations (Bryden et al., 2011; Garzoli



et al., 2013) of an indicator of bistability (de Vries and Weber, 2005; Dijkstra, 2007) suggest that the present-day AMOC is in

such a bistable regime.

The collapse of the AMOC is thought to be a rare event but because of its high impact, it is important to compute the probability of its occurrence in the $21^{st}$ century. The theoretical framework of Large Deviation Theory (Freidlin and Wentzell, 1998) is not applicable as strong assumptions on the noise statistics have to be made. Moreover, the methods from this theory prove unfeasible in high-dimensional systems, such as global ocean models. Transition probabilities can also be computed

using a simple Monte-Carlo approach in which many long trajectories are simulated to find enough transitions to determine statistics. However, this approach is not tractable either for high-dimensional systems because of the required computational costs.

A good alternative is to use splitting, or cloning algorithms, such as Trajectory-Adaptive Multilevel Splitting (TAMS) (Lestang et al., 2018; Baars et al., 2021) based itself on the AMS (Adaptive Multilevel Splitting) algorithm (Cérou and Guyader,

2007). Using ensemble simulations (with significantly fewer members compared to a traditional Monte-Carlo approach), these methods are suited to compute the probability to reach a state $B$ (e.g., collapsed AMOC) of the phase space starting from a state $A$ (e.g., present-day-like AMOC), where the transition from $A$ to $B$ is a rare event. TAMS also adds a time threshold: the transition must be completed before a certain time $T_{max}$. Using a score function to measure how close trajectories are from the state $B$, AMS and TAMS encourage the closest ones while discarding the ones where the rare event is least prone to happen.

Then, new trajectories are simulated by branching from the most promising ones. In this way, the statistics are not altered and the transition probability can be obtained at a lower computational cost. Rolland et al. (2015) applied AMS for the first time to the computation of rare-event probabilities in a 1-D stochastic partial differential equation. Since then, such sampling algorithms have achieved numerous successes when applied to atmospheric turbulent flows (Bouchet et al., 2019; Simonnet et al., 2021) or even a full complexity climate model (Ragone et al., 2018).

The main limitation of this kind of algorithms is that they heavily rely on their score function; using a bad score function may cancel its time-saving benefit. Fortunately, in the case of TAMS, the optimal score function is known: it is the committor function $q$ (Cérou et al., 2019). It is defined as the probability to reach a certain set $A$ of the phase space before another set $B$, as a function of the initial condition of the trajectory. The committor function is a solution of a backward Kolmogorov equation and, in theory, it is possible to compute it exactly. In practice, however, this equation is intractable to solve in high-dimensional

models. Even in simpler systems, such as the Jin-Timmermann model, Lucente et al. (2022a) showed that the committor function can already have a very complex structure. Moreover, the committor function contains precisely the information we are looking for when using TAMS. Consequently, another way to estimate that function is required.

When it is assumed that the underlying dynamics can be described by an overdamped Langevin equation, the backward Kolmogorov equation may be simplified. The committor can then be parametrized using, for instance, feedforward neural

networks (Khoo et al., 2018; Li et al., 2019). A novel way to perform such parametrization has recently been proposed by Chen et al. (2023) by using Tensor Networks. More general approaches have also been developed to estimate the committor from pre-computed trajectories. For instance, Milestoning (Elber et al., 2017) consists in coarsening the phase space into a cell grid and considering short trajectories between boundaries of these cells. Lucente et al. (2022b) has also recently applied





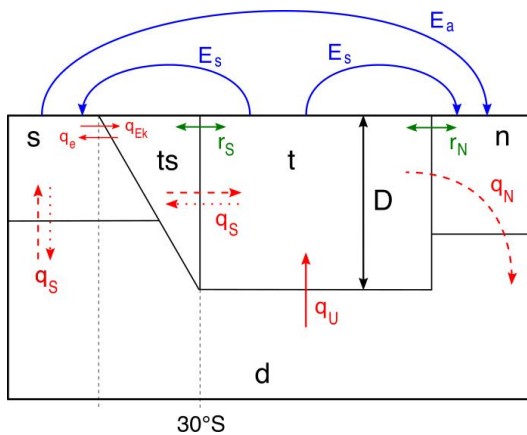

**Figure 1.** Sketch of the AMOC box model (Castellana et al., 2019). The red arrows represent the volume transport between each box, the blue arrows the freshwater forcing and the green arrows the wind-driven transports. Solid red arrows are transports that always take place, dashed arrows are only present in a present-day-like AMOC and dotted arrows in its reversed state.

to this problem a Markov-chain interpretation of the Analogue method (Lorenz, 1969a, b; Yiou, 2014; Yiou and Déandréis, 2019; Lguensat et al., 2017; Platzer et al., 2021a, b), which consists in approximating the dynamics of the system by sampling its phase space. The resulting simpler process is a transition matrix which properties can be easily studied. The committor estimation problem was also approached by trying to solve the backward Kolmogorov equation using a data-driven mode decomposition of the adjoint of the Fokker-Planck operator (Thiede et al., 2019; Strahan et al., 2021; Finkel et al., 2021). Finally, neural network methods can also be applied for direct computation of the committor function in this more general setting, as was shown by Lucente et al. (2019) and recently developed by Miloshevich et al. (2022).

The contribution of the present paper is to compare the capabilities and performance of these different committor estimation methods by applying them to two different conceptual, low-dimensional, ocean models. The objective is to assess their strengths and weaknesses and determine which one could be best suited for applying the committor estimation within TAMS for high-dimensional models. The structure of the paper is as follows. In Sect. 2, we shortly present both ocean models for which we estimate the committor. We also explain the methods that will be compared, detail the choices made for their implementation and outline our comparison protocol. Results on the performance of the different committor estimation methods are presented in Sect. 3. In Sect. 4, we discuss possible ways of optimization and future lines of improvement.

## 2 Models and methods

### 2.1 The AMOC model

The AMOC box-model used here was presented in Cimatoribus et al. (2014) and slightly extended in Castellana et al. (2019). The Atlantic Ocean is divided into five boxes as shown in Fig. 1. The Northern and Southern Atlantic boxes are labelled $n$ and





$s$, respectively. The pycnocline layer is modelled as two boxes, the Tropical box ($t$) and the Tropical Southern box ($ts$), the latter located south of $30°$S. Finally, a Deep box ($d$) extends throughout the whole ocean below the pycnocline depth $D$. The temperature in each box is prescribed, so that the water density only depends on its salinity. Due to conservation of salt, the

state vector of the model is determined by 5 of the 6 quantities $S_t$, $S_{ts}$, $S_n$, $S_s$, $S_d$ and $D$.

The flows between the boxes are represented by three main quantities. Firstly, the volume transport $q_n$ accounts for the downwelling of dense water taking place in the Northern box. Secondly, $q_s$ corresponds to the difference between the wind-driven Ekman flow ($q_{Ek}$) and the eddy-induced flow ($q_e$). Finally, $q_u$ models the upwelling from the Deep box to the Tropical box. Two additional terms, $r_s$ and $r_n$, represent the salinity transport due to the wind-driven subtropical gyres. The equations

of the model are given in the Appendix A.

The model is subject to two forced freshwater fluxes: a constant symmetric forcing, $E_s$, from the Tropical box to the boxes $n$ and $s$ and an asymmetric forcing $E_a$ from the box $s$ to the box $n$. Only $E_a$ contains a random white noise component, i.e. $E_a(t) = \overline{E_a}(1 + f_\sigma \zeta(t))$, where $\zeta(t)$ is a white noise process with zero mean and unit variance. The quantity $f_\sigma$ is the noise ratio ranging from $0$ to $0.5$ (Castellana et al., 2019). Fixing all other parameters (see Appendix A), the behaviour of the

trajectories of the system is entirely determined by the values of the two parameters $(\overline{E_a}, f_\sigma)$.

## 2.2    The double-gyre model

The double-gyre model is a well-studied model of the wind-driven ocean circulation in a rectangular basin of size $L_x \times L_y$ and characteristic horizontal scale $L$. This model describes the flow in an ocean layer of constant density and fixed thickness, forced by an idealised zonal wind stress. The dimensionless equation for the geostrophic stream function $\Psi$ and the potential

vorticity $Q$ are given by

$$Q \equiv \nabla^2 \Psi - \Psi + \beta y \tag{1}$$

$$\frac{\partial Q}{\partial t} + J(\Psi, Q) - r\nabla^2 \Psi = \sigma(1 + \gamma\zeta(t)) \sin\left(2\pi \frac{L}{L_y} y\right) \tag{2}$$

where $J$ is the Jacobian $J(u, v) = \dfrac{\partial u}{\partial x}\dfrac{\partial v}{\partial y} - \dfrac{\partial u}{\partial y}\dfrac{\partial v}{\partial x}$. $\beta$ stands for the strength of the planetary vorticity gradient and $r$ for the bottom friction coefficient. The wind-stress forcing amplitude is $\sigma(1 + \gamma\zeta(t))$, where $\zeta(t)$ is again a zero-mean unit-variance

white-noise process, $\sigma$ is the deterministic strength of the wind-stress and $\gamma$ is a stochastic noise ratio. In deriving a reduced model, a Fourier expansion (with a western-boundary layer structure) is pursued (Jiang et al., 1995), i.e.,

$$\Psi(x, y, t) = \sum_{k=1}^{4} A_k(t) e^{-sx} \sin(x) \sin(ky) \tag{3}$$

where $s$ represents the fixed width of the boundary layer. In a Galerkin method, the equations are projected onto the same Fourier basis. As shown in the Appendix B, this truncation gives four ODEs that capture (Simonnet et al., 2005) the first

bifurcations of the full model (Eq. 2). When all other parameters are fixed (values given in Appendix B), the behavior of the model is fully determined by the values of the two parameters $(\sigma, \gamma)$.



## 2.3 Committor function

Consider two sets $A$ and $B$ in the phase space $\Omega$ of a given dynamical system. From a trajectory $X(t)$ of that dynamical system, one can define its first-passage time $\tau_C$ in any set $C \in \Omega$ as:

$$\tau_C = \min\{t \mid X(t) \in C\} \tag{4}$$

The committor function, $q(x)$, is the probability that a trajectory $X(t)$ starting at $x$ reaches the set $A \in \Omega$ before the set $B \in \Omega$, i.e.,

$$q(x) = \mathbb{P}(\tau_A < \tau_B \mid X(0) = x) \tag{5}$$

In the case of a stochastic dynamical system with dimension $d$ (which is the case we always consider here), the committor function is a solution to the backward Kolmogorov equation. In other words, it obeys the following Dirichlet problem:

$$\mathcal{L}q(x) = 0 \; \forall x \in (A \cup B)^c \tag{6}$$

$$q(x) = 1 \; \forall x \in A \tag{7}$$

$$q(x) = 0 \; \forall x \in B \tag{8}$$

where $\mathcal{L}$ is the infinitesimal generator of the process and the adjoint of the Fokker-Planck operator. In the case considered here of a stochastic differential equation, it is defined as:

$$\mathcal{L} = \sum_{i=1}^{d} a_i(x) \frac{\partial}{\partial x_i}(\cdot) + \sum_{i,j=1}^{d} D_{ij}(x) \frac{\partial}{\partial x_i \partial x_j}(\cdot), \tag{9}$$

$a$ being the drift of the system and $D$ its diffusivity.

One can directly sample the committor function via a Monte-Carlo method. Suppose we have determined a trajectory $X(t)$ and we want to compute the committor for each point $x \in \Omega$ in this trajectory. Then, for each $x$, $N$ trajectories starting from $x$ are simulated. If $N_A$ is the number of those where $\tau_A < \tau_B$, the committor is simply expressed as:

$$q(x) = \frac{N_A}{N} \tag{10}$$

Such a Monte-Carlo computation, however, is extremely costly, even when fully optimized and parallelized. This method is only used here to obtain a reliable "true" committor for comparison with their estimated equivalent.

## 2.4 Committor estimation: Analogue Methods (AMC)

The Analogue method was first proposed in Lorenz (1969a, b) as a way to predict future states in a trajectory by using past data. Much work has been done on this method (Yiou, 2014; Yiou and Déandréis, 2019; Lguensat et al., 2017; Platzer et al., 2021a, b) and it has been used to generate new stochastic trajectories by re-using past data to emulate the dynamics of the system. This method may also be interpreted as a Markov Chain that approximates the underlying dynamics. This interpretation allows to



store an effective dynamics in a simple transition matrix, that Lucente et al. (2022b) used to compute a committor function. We
give below a short summary of this method.

Let $\{X_n\}_{1\leq n\leq N}$ be a discretized trajectory. Each state $X_n$ corresponds to a simulated time $t_n = n\Delta t$, $\Delta t$ being the time resolution. The analogues of every state $X_n$ are defined as its $K$ nearest neighbours $\{X_n^{(k)}\}_{1\leq k\leq K}$ in the trajectory, using the Euclidean distance in the phase space. In practice, the analogues are obtained through a search in a KD-tree (Bentley, 1975) containing every point of $\{X_n\}_{1\leq n\leq N}$. KD-trees are a type of binary space-partitioning trees: every node of the tree
corresponds to a k-dimensional data point and belongs to a hyperrectangle splitting the space along a certain axis. This type of trees is well suited to search for nearest neighbours efficiently.

This set of analogues is thus a subset of the original trajectory: $\{X_{l_k} \mid l_k \in [1, N-1] \setminus \{n\}, 1 \leq k \leq K\}$. It is then assumed that there is a transition from $X_n$ to the image of any of its $K$ analogues with a probability $1/K$. Hence, the end-point of the trajectory $X_N$ is excluded from the set of analogues because it has no image. $X_n$ has thus equal probability to transition to
any of the states $\{X_{l_k+1}, l_k \in [1, N-1] \setminus \{n\}, 1 \leq k \leq K\}$. A Markov chain can then be built that approximates the dynamical behavior of the original trajectory. The transition matrix $G$ describing this Markov chain has elements

$$G_{i,j} = \begin{cases} \frac{1}{K} & \text{if } X_{j-1} \text{ is an analogue of } X_i \\ 0 & \text{otherwise} \end{cases} \tag{11}$$

Suppose that transitions occur between two sets of the phase space, called $A$ and $B$. Firstly, all states belonging to $A$ in the trajectory can be grouped together. The same is done for the states belonging to $B$. Then a new transition matrix $\tilde{G}$ can be defined where all states in $A$ and $B$ are represented by a single index, respectively $i_A$ and $i_B$. The elements of $\tilde{G}$ are then

$$\begin{cases} \tilde{G}_{i_A,i_A} = 1 \\ \tilde{G}_{i_B,i_B} = 1 \\ \tilde{G}_{i_A,j} = 0 & \text{if } j \neq i_A \\ \tilde{G}_{i_B,j} = 0 & \text{if } j \neq i_B \\ \tilde{G}_{i,i_A} = \sum_{k|X_k \in A} G_{ik} & \text{if } i \neq i_A, \, i \neq i_B \\ \tilde{G}_{i,i_B} = \sum_{k|X_k \in B} G_{ik} & \text{if } i \neq i_A, \, i \neq i_B \\ \tilde{G}_{i,j} = G_{i,j} & \text{otherwise} \end{cases}$$

The committor function is now computed from this transition matrix (Lucente, 2021). Let $\mathbf{q}$ be the vector containing the committor function $q(x)$ on every point of the trajectory. It follows (Schütte et al., 1999; Prinz et al., 2011; Tantet et al., 2015;
Noé and Rosta, 2019; Lucente, 2021) that $\mathbf{q}$ obeys the following equation

$$\tilde{G}\mathbf{q} = \mathbf{q}. \tag{12}$$

Finding the committor function on the trajectory $\{X_n\}_{1\leq n\leq N}$ thus amounts to solving an eigenvector problem. It can be shown that $\tilde{G}$ has an eigenvalue 1 with two leading eigenvectors $\mathbf{v}_1$ and $\mathbf{v}_2$ (Prinz et al., 2011). The committor function then reads:

$$\mathbf{q} = \alpha\mathbf{v}_1 + \beta\mathbf{v}_2 \tag{13}$$





where $\alpha$ and $\beta$ are derived from the conditions $q_{i_B} = 0$ and $q_{i_A} = 1$ (following the same convention as in Sect. 2.3).

AMC thus returns an estimate of the committor on every point of the input trajectory. Since all the information only comes from the transition matrix computed from that trajectory, AMC does not require any pre-training (in contrast to the machine learning methods described below) and could in theory be applied directly to any trajectory computed in TAMS. However, restarting the whole process from scratch for each of the hundreds of trajectories simulated during TAMS may be computa-

tionally expensive.

In order to estimate the committor at any other point of the phase space (not belonging to the train trajectory), the AMC method has to be combined with another method. As suggested in Lucente (2021), we use a K-nearest neighbours (KNN) method (Altman, 1992). More precisely, we will use AMC to train the Markov Chain and KNN to apply the learnt Markov Chain to test trajectories.

Suppose an estimate $q$ of the committor is already known on a set of states $\{X_n\}_{1 \leq n \leq N}$. To compute the committor on another point $Y$ of the phase space, its K-nearest neighbours $\{X_{l_k}, l_k \in [1, N], 1 \leq k \leq K\}$ can be computed using again the Euclidean distance in the phase space. The committor on $Y$ is then calculated from

$$q(Y) = \frac{1}{K} \sum_{k=1}^{K} q(X_{l_k}) \tag{14}$$

In practice, AMC will be applied beforehand on a very long training trajectory to create a pool of states for which the committor

is already estimated. Another KD-tree is then used to computed the K-nearest neighbours of $Y$ among this pool of states. For simplicity, we use the same value of $K$ for both the number of analogues used in AMC and the number of nearest neighbours used in KNN.

When applying AMC, as explained in Lucente (2021), there are cases where the estimated committor takes values outside of the interval $[0, 1]$. Such cases can occur when the dataset is not large enough, thus causing a breakdown of ergodicity in the

Markov chain. In practice (Lucente, 2021), the pool of states created by AMC only consists of the points $x$ where $q(x) \in [0, 1]$.

## 2.5 Committor estimation: Neural Network methods

### 2.5.1 Feedforward Neural Network (FFNN)

FFNN's are frequently used to perform a classification task: each data sample is labelled (often with a binary label) as belonging to a class and the FFNN must learn the different classes. Here, however, to estimate a probability, an extra layer shall be added

at the end of the network. First, all data samples must be labelled in both the train and test set. Following hte same convention as in Sect. 2.3, two classes are used: "leading to a state in $A$" and "leading to a state in $B$". Let $X$ be a point belonging to a given trajectory. Starting from $X$, if that trajectory then first reaches a state in $A$, $X$ is labelled as "leading to a state in $A$" and is assigned the label $(0, 1)$. Otherwise, if after going through $X$, the trajectory first reaches a state in $B$, $X$ will be labelled as "leading to a state in $B$" and will be assigned the label $(1, 0)$. Classes are here labelled using one-hot encoding as it will allow

to transform these labels into a probability. In this way, all data samples (in the train and test sets) are in the form: $\{X, Y\}$ where $X$ is a point in the phase space and $Y$ is either $(0, 1)$ or $(1, 0)$.





The FFNN itself consists of several hidden layers of densely connected neurons, preceded by an input layer and followed by an output layer. Our baseline architecture contains 3 hidden layers of respectively $64, 128$ and $256$ neurons. The size of the input layer is the number of variables given as input, between $2$ and $4$. The output layer always contains $2$ neurons, so as to predict one-hot-encoded labels. This kind of labels are used because they allow to apply a Softmax function on the output of the FFNN. This function transforms the output of the network into a couple of probabilities, that can be interpreted as: ("probability to reach a state in $B$ first", "probability to reach a state in $A$ first"). The second member of that couple is the desired committor function.

The loss function used to train the FFNN is the cross-entropy loss. It is well suited to assess a distance between the true committor and the data-based estimation. Moreover, it is closely related to the measures we are using to evaluate the performance of the different methods (see Sect. 2.7.1).

Unlike the training of AMC, training a FFNN involves randomness (e.g., shuffling of the train dataset). To ensure robustness of the results, we use $k$-fold cross-validation during the training process. It consists in randomly splitting the whole training set into $k$ subsets and then training the FFNN $k$ times. Each time, validation is performed using a different subset and the remaining $k-1$ are used for training. This method allows to make statistics on the performance of the network for a given setup. Here, we arbitrarily choose $k = 20$.

The AMC method is easy to optimise as it involves a single hyperparameter. However, for the FFNN, there are many more parameters that can be varied. We choose the following setup and hyperparameters:

– Each layer of the neural network is initialised according to the He et al. (2015) normal initialisation method.

– The optimisation algorithm is the Stochastic Gradient Descent method.

– We use a learning rate scheduler, with the plateau algorithm: if the loss function is not improved for $5$ consecutive epochs, the learning rate is divided by $10$; the initial learning rate $\lambda = 10^{-4}$.

– Each learning lasts $30$ epochs. At the end, we retain the state of the model at the epoch that resulted in the best validation loss.

### 2.5.2 Reservoir Computing (RC)

Reservoir Computing was first introduced by Jaeger (2001). It is a specific type of Recurrent Neural Networks (RNN). The main difference between the latter and the FFNN's presented in the former section is that the connection structure of RNN contain cycles. In this way, they are able to preserve a dynamical memory of their internal state, making RNN a powerful tool for dynamical system analysis. However, training traditional RNN with a gradient-descent algorithm suffers from a number of drawbacks that make it inefficient (for more details, see the introduction of Lukoševičius and Jaeger (2009)). Reservoir Computing avoids those problems by using a structure that does not require gradient-descent.

A classical Reservoir Computer consists of three main elements: an input layer matrix $\mathbf{W}_{in}$, a random network (the reservoir) with the reservoir state $\mathbf{X}$ and an output layer matrix $\mathbf{W}_{out}$. The main feature of this method is that the weights of the input





layer and reservoir are fixed: only the output layer is trained, using a regularized linear least-squares method. In a nutshell, the
input time series is mapped onto the reservoir with a nonlinear function (usually $\tanh$); the output layer then simply performs
a linear regression of the feature vector $\mathbf{X}$ computed in the reservoir. It has recently been shown (Gonon and Ortega, 2020) that
a universal approximator can be realized with this approach. However, this classical approach has again several drawbacks,
in particular the use of random networks to represent the reservoir and the number (7) of hyperparameters that have to be
optimized; both can greatly hinder the performance of the Reservoir Computer.

Recently, Gauthier et al. (2021) developed the so-called "Next-Generation" Reservoir Computing, which we abbreviate
below with RC. The basic idea behind RC is that, instead of first applying a nonlinear function to the data followed by a linear
regression as in the classical approach ; first a linear function is applied to the data and then the output layer is a weighted sum
of nonlinear functions. By doing so, no more reservoir is needed. The details of the methodology are described in Gauthier
et al. (2021), and we only specify below the relevant ones for the committor estimation.

Consider a trajectory $\boldsymbol{U} \in \mathbb{R}^{M \times T}$ in dimension $M$ and consisting of $T$ time steps. From this trajectory a feature vector
$\mathbf{X} \in \mathbb{R}^{N \times (T-ks)}$ is extracted. $N, k$ and $s$ are detailed in Appendix C. The output layer then simply maps this feature vector $\mathbf{X}$
onto the desired output. The output layer is represented by a matrix $\mathbf{W}_{out} \in \mathbb{R}^{D \times N}$ where $D$ is the dimension of the desired
output. In our case, $M$ is the number of variables in the model where we aim to estimate the committor and $D = 1$ (dimension
of the committor). The RC method returns the committor via

$$\boldsymbol{q} = \mathbf{W}_{out}\boldsymbol{X} \tag{15}$$

Suppose we have computed the full time series of feature vectors $\boldsymbol{X}$. Then, if we know the committor $\boldsymbol{q}_{train}$ of the training
set, $\boldsymbol{W}_{out}$ is given by

$$\mathbf{W}_{out} = \boldsymbol{q}_{train}\boldsymbol{X}^{\mathrm{T}}(\boldsymbol{X}\boldsymbol{X}^{\mathrm{T}} + \alpha\mathbf{I})^{-1} \tag{16}$$

where $\mathbf{I}$ is the identity matrix and $\alpha$ is the Tikhonov regularization parameter.

Appendix C explains how the feature vector $\boldsymbol{X}$ is determined from values of $\boldsymbol{U}$ at $k$ time steps with stride $s$. This RC method
only depends on four hyperparameters, which are $\alpha$, $k$ and $s$ the degree $p$ of the monomials in the nonlinear part of $\mathbf{X}$ (see
Appendix C). These are very convenient to optimize because $k$, $s$ and $p$ are integers, which must all remain small to keep
tractable computation times. The values of all hyperparameters are empirically determined, model-dependent and are listed in
Table C1 in Appendix C.

## 2.6 Dynamical Galerkin Approximation (DGA)

The Dynamical Galerkin Approximation (DGA) method as implemented here is based on Thiede et al. (2019) and Finkel et al.
(2021). The main idea is to project the Dirichlet problem (Eq. 6) onto a set of basis functions estimated from data. The original
problem is thus reduced to a simple matrix equation that gives access to the projection of the committor function onto this
basis.





250   The first step is to homogenise the boundary conditions in Eq. (6). To do so, this system is rewritten in terms of a function $g(x) = q(x) - \mathbb{1}_B(x)$, where $\mathbb{1}_B(x)$ is the indicator function of the set $B$, i.e.

$$\mathbb{1}_{\mathbb{B}}(x) = \begin{cases} 1 \text{ if } x \in B \\ 0 \text{ otherwise} \end{cases} \tag{17}$$

The original Dirichlet problem now reads:

$$\mathcal{L}g(x) = -\mathcal{L}\mathbb{1}_{\mathbb{B}}(x) \;\forall x \in (A \cup B)^c \tag{18}$$

255   $$g(x) = 0 \;\forall x \in (A \cup B) \tag{19}$$

Next, a set of basis functions $\{\phi_i, \ i \in [1, M]\}$ is defined within the Hilbert space on which Eq. (18-19) are defined. The key constraint is that each basis function should obey the homogeneous boundary conditions. It ensures that the projection of $g$ onto this subspace also obeys the boundary conditions. Calling L the projection of $\mathcal{L}$ onto this subspace and $\bar{g}$ that of $g$, each basis function must then obey:

260   $$\langle \phi_i, \mathrm{L}\bar{g} \rangle = -\langle \phi_i, \mathrm{L}\mathbb{1}_{\mathbb{B}} \rangle \tag{20}$$

Furthermore, for two time series $u$ and $v$ of size $N$ and a time step $\Delta t$, the definition of the generator $\mathcal{L}$ gives:

$$\langle u, \mathcal{L}v \rangle = \frac{1}{N} \sum_{i=1}^{N-1} u_i \frac{v_{i+1} - v_i}{\Delta t} \tag{21}$$

By construction, there is a unique set of scalars $a_j$ such that:

$$\bar{g}(x) = \sum_{j=1}^{M} a_j \phi_j(x) \tag{22}$$

265   By writing $L_{ij} = \langle \phi_i, \mathrm{L}\phi_j \rangle$ and $r_i = \langle \phi_i, \mathrm{L}\mathbb{1}_{\mathbb{B}} \rangle$, all that remains is the matrix equation:

$$\sum_{j=1}^{M} L_{ij} a_j = -r_i \tag{23}$$

The main difficulty with this approach is to find a good set of basis functions but Thiede et al. (2019) also provides a method to find these functions (see Appendix D).

In practice, the modes are computed from a training set. Then, they are extended on the trajectories where the committor is

270   to be estimated using an approximation formula provided by Thiede et al. (2019) (see Appendix D).

## 2.7   Performance evaluation

We are not only looking for a method that best estimates the committor function, but also for one that is most time efficient. We will therefore use several measures to compare them: the logarithm score, the difference score and the computation time. In this section, we give more details about the first two.





### 2.7.1 Logarithm score

Let $\{x_k\}_{k \in [1,N]}$ be a trajectory of length $N$ and $\{q_k\}_{k \in [1,N]}$ an estimate of the corresponding committor. To every state $x_k$, a label $z_k \in \{0,1\}$ can be attached such that $z_k = 1$ if $x_k$ leads (in the trajectory) to an off-state and $z_k = 0$ if $x_k$ leads to an on-state. If $x_k$ itself is an on-state or an off-state, it is naturally labelled 0 or 1 respectively. If $N_l$ is the index of the last state in the trajectory belonging to either the on-zone or the off-zone, then the last $N - N_l$ states in the trajectory cannot be labelled. Hence, these are not included in the computation of the logarithm score.

The logarithm score is defined (Benedetti, 2010) as:

$$L = \frac{1}{N_l} \sum_{k=1}^{N_l} (z_k \ln(q_k) + (1 - z_k) \ln(1 - q_k)) \tag{24}$$

This score has values between $-\infty$ and 0, where $L = 0$ corresponds to a perfect agreement between the theory and the estimation, while $L = -\infty$ corresponds to the "opposite match".

Let $X$ be a state in the phase space. If $X$ is not an on-state nor an off-state, its committor value is $0 < q(X) < 1$ in the general case. This is the probability that a trajectory starting on $X$ reaches an off-state before an on-state. However, all we have access to is a realization of this event. If we look at the available data, $X$ either leads to an off-state or does not. As a result, from the logarithm score's viewpoint, the committor function is always either 0 or 1. The true committor obtained by Monte-Carlo sampling has thus a "real score" $L_{MC} < 0$. $L_{MC}$ is the value we should aim for when estimating the committor. In other words, a perfect estimate of the committor will have a logarithm score $L_{MC}$. However, it is not guaranteed that an estimate of the committor with a score $L_{MC}$ is the perfect estimate.

For better interpretability, (Benedetti, 2010) also defines the normalized logarithm score, as:

$$S = 1 + \frac{1}{C_S} \frac{1}{N_l} \sum_{k=1}^{N_l} (z_k \ln(q_k) + (1 - z_k) \ln(1 - q_k)) \tag{25}$$

where $C_S$ is a reference term, the climatology. The climatology is defined here as the score we obtain if we predict everywhere the average of a reference committor. For a meaningful comparison, the on and off-states of the reference committor are excluded from the average. Calling $\langle q \rangle$ the average reference committor over the transition states, $C_S = -\langle q \rangle \ln(\langle q \rangle)$. A normalized logarithm score of $S = 0$ is equivalent to predicting everywhere the climatology. The theoretical "perfect match" corresponds this time to a normalized score of $S = 1$. This normalized score is still not bounded below 0 (so the "opposite match" still corresponds to $S = -\infty$).

### 2.7.2 Difference score

The difference score is simply the squared difference between the estimated committor (called $E$) and the true one (called $T$). Moreover, we use $F$ as the "furthest estimate" of the true committor. It consists in rounding every value of the committor to the furthest integer, either 0 or 1, so as to maximize the quantity $||T - F||^2$. The difference score $D$ is then defined as:

$$D = 1 - \frac{||T - E||^2}{||T - F||^2} \tag{26}$$





This score has two big advantages. Firstly, it is very easy to interpret. A score $D = 1$ corresponds to predicting exactly the true committor while a score $D = 0$ corresponds to the worst possible estimate. Since the climatology here corresponds to the mean committor of a reference committor, $D = 0.5$ roughly corresponds to predicting the climatology. So, unlike the normalized logarithm score, a difference score closer to 1 is always a better estimate of the committor.

The major drawback of using the difference score is that it is in general not computable. Indeed, in the general case, we do

not know the true committor. In this paper, however, thanks to the low-dimensionality of our example models, we can determine the true committor with a Monte-Carlo method. We can use this score here in the comparison of the different methods but in more complex settings we will have to rely entirely on the normalized logarithm score.

## 3    Results

We will now compare the different methods used to estimate the committor on both ocean models. Assessing the measures of

the committor estimate and the computation time for each method enables us to assess which one is the best and seems the most promising for future applications, on high-dimensional models in particular. All results presented below were computed on a Mac M1 CPU using Python 3.9.7, NumPy 1.22.3 and PyTorch 1.10.2 (the latter only for the FeedForward Neural Network). Computation times are simply the elapsed time during the training/testing process.

### 3.1    AMOC model

#### 3.1.1    Phase space analysis

As detailed in Castellana et al. (2019), the AMOC model exhibits a bistability regime for $\overline{E_a} \in [0.06, 0.35]$. Its two stable steady states are defined by $q_n > q_s > q_u$ and by $q_n = 0$ & $q_s < 0$. The former corresponds to a strong downwelling in the northern Atlantic, and thus to the present-day circulation of the AMOC. The latter corresponds to the fully collapsed state of the AMOC, with a shut down of the downwelling in the northern Atlantic and a reversal of the southern circulation. These definitions are

actually more general than the fixed points of the system: they define entire sets of the phase space, to which the fixed points respectively belong. So, here, what we call an "on" state of the AMOC is any point such that $q_n > q_s > q_u$, not only a fixed point of the system. When noise is added to the system ($f_\sigma > 0$), a second type of collapse is observed: $q_n = 0$ & $q_s > 0$. In this case, fast variations in the freshwater inputs may shut down the northern downwelling without disturbing the deep layers of the ocean. This shut-down is always a transient state of the AMOC and happens on much shorter time scales than a full

AMOC collapse with an adjustment of the deep ocean circulation ($q_s < 0$). With the values of $\bar{E}_a$ and $f_\sigma$ considered here, a 'temporary' shut-down occurs after a few decades to a century, while the transition to the fully collapsed state takes about 1000 years. Here, we are interested in short-term transition probabilities of the AMOC, hence we focus on the transient collapse, which we call an "off" state of the AMOC. Hence, we will study transitions between "on" and "off" states.

The expressions of $q_s$ and $q_u$ only depend on $D$, while $q_n$ is entirely defined by $S_n - S_{ts}$ and $D$. Consequently, we can

summarise the behaviour of the system in the reduced phase space $(S_n - S_{ts}, D)$, shown in Fig. 2a for $f_\sigma = 0.4$ and $\bar{E}_a = 0.234$.





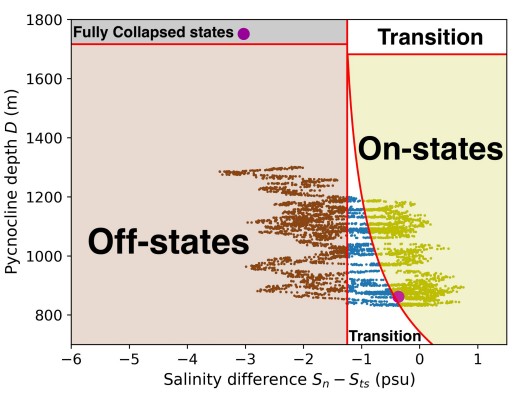
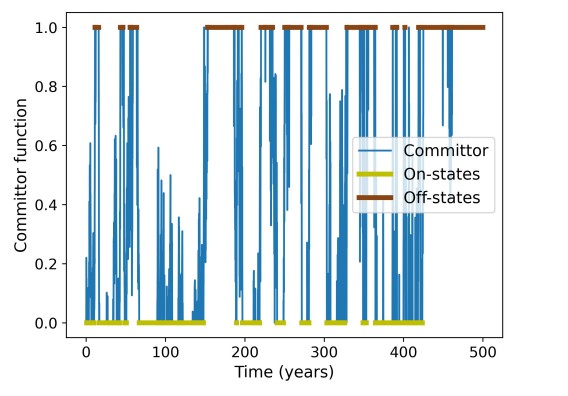

(a)                                                                                              (b)

**Figure 2.** (a) An example of a long trajectory in the reduced phase space $(S_n - S_{ts}, D)$, for $(\bar{E}_a, f_\sigma) = (0.234, 0.4)$. The phase space contains 4 zones: on-states (yellow), off-states (brown), fully-collapsed states (grey) and a transition zone (white). The fuchsia points are the steady states. The red curves show the separation between each zone. (b) The corresponding committor function. The off-states are highlighted in brown and the on-states are highlighted in yellow.

In this figure, the different zones of interest in the phase space are highlighted: yellow for the set containing all "on" states, brown for the set containing all "off" states and black for the set containing all fully-collapsed states. The purple dots represent the steady states. We also plot a long trajectory starting close to the steady on-state. We clearly see the transitions between on-states and off-states, or F-transitions (fast transitions) (Castellana et al., 2019), through a transition zone. In this example,

the simulated trajectory is too short for the system to reach the fully collapsed zone, or to undergo a so-called S-transition (slow transition) (Castellana et al., 2019).

We are interested in the probability that the AMOC collapses, so the committor function is here defined as the probability that a trajectory reaches the off-states zone before the on-states zone. Here, the subspace $A$ is the set of all on-states of the AMOC (yellow zone on Fig. 2a) and the subspace $B$ is the set of all off states of the AMOC (brown zone on Fig. 2a). So, here,

all on-states will correspond to a value of the committor $q(x) = 0$ and all off-states will correspond to a value of the committor $q(x) = 1$. In such a setting with fixed forcing, fixed noise and a time-independent phase space, there is a one-to-one mapping between any trajectory in the phase space and its corresponding committor function. We can thus plot in Fig. 2b the committor along the simulated trajectory of Fig. 2a. The committor function is plotted in blue. The on-states, with a committor value of $0$, are highlighted in yellow while the off-states, with a committor value of $1$, are highlighted in brown. The committor function

switches between both kinds of states, which corresponds to the noise-induced transitions in the original trajectory.

In the following study of this model, all trajectories will be computed with $(\bar{E}_a, f_\sigma) = (0.234, 0.4)$.





### 3.1.2 Train and test dataset

To be able to compare different methods for the committor estimation, we need to train them and to test them with different
trajectories. We thus need to create a training and a test dataset. For simplicity and consistency, we set a standard length for
all test trajectories of both models: 5000 time steps (corresponding to 500 years of simulation, similar to the trajectory and its
associated committor function plotted in Fig. 2). The total test set consists of 100 independent such trajectories. The logarithm
and difference scores are averaged over these trajectories.

When it comes to training methods for the committor estimation problem, what really matters is to have reactive trajectories,
that is, trajectories going through both on and off-states. Consequently, it makes sense to count the length of trajectories in
terms of the number of transitions $N_T$. What we call a transition is a set of consecutive points starting in the on-zone (resp.
off-zone) and ending in the off-zone (resp.on-zone).

One of our goals is to estimate the committor function using as less data as possible. It is thus interesting to study how
the performance of each method scales with the amount of training data. To do so, we generate several training sets, having
an increasing number of transitions $N_T$ that spans a large interval: $N_T = 10, 20, 30, 50, 75, 100, 150, 200, 300, 400, 500$. In
practice, to ensure meaningful comparison between these training sets, we only generate the longest one, with $N_T = 500$. The
shorter ones simply consist of the $N_T$ first transitions of this very long trajectory.

In the case of the AMOC model, generating a trajectory with 500 transitions is not possible without S-transitions to occur,
which we want to avoid. Instead, we concatenate as many 5000 timestep long trajectories as needed, all starting close to the
steady on-state.

### 3.1.3 Application of the different methods

The performance of each method when applied to the AMOC model will be presented in the next two sections but first, we
specify some implementation details.

The training of AMC, DGA and RC does not involve randomness, hence we perform it once only on the entire training
set containing $N_T$ transitions. As for FFNN, since the initialization of its weights and the gradient descent algorithm involve
randomness, we perform a 20-fold cross validation on the same training set with $N_T$ transitions. The latter is thus split into 20
subsets and only 19 are used for training every time. In the end, we average the performance obtained from the resulting 20
optimized FFNN.

In the case of RC, the parameters $k$, $s$, $p$ and $\alpha$ (see Appendix C) were optimized by hand and their values are in Table C1.
In the case of DGA, the number of modes, the values of $d$, $\epsilon_0$ and $\epsilon$ (see Appendix D) were also optimized empirically (see
Table D1).

For optimal results, the different methods are also applied to different sets of variables , which shows that the methods
capture different features of the phase space:

- AMC and DGA : $S_n - S_{ts}, S_n$ and $S_s$

- FFNN: $S_n - S_{ts}$ and $D$





**Figure 3.** Comparison of the performance and computation time of all four methods on the AMOC box model.

(a) The normalized logarithm scores of each method.

(b) The difference scores of each method.

(c) The training times of each method.

(d) The testing times of each method.

In (a), (b) and (d), each curve is the average over the score of the 100 test trajectories. The shaded areas around each curve are their 90% most probable values, between the 5th and 95th percentiles. In (c), the shaded area is only provided for FFNN, since all other methods are only trained once.

– RC: $S_n - S_{ts}, S_n, S_s$ and $D$

### 3.1.4   Skill

The normalized logarithm score for each method is presented in Fig. 3a. Firstly, AMC (blue curve) and DGA (green curve) both show a great performance, since they are the two best-performing methods up to $N_T = 100$. The lowest score of AMC,





for $N_T = 10$, is $0.660$. Its best normalized logarithm score is attained for $N_T = 500$, where the score is $0.718$. The gap between
its 5th and 95th percentile does not decrease in the same time and remains between $0.299$ and $0.331$ (slightly higher than the
width of the distribution of the true score, equal to $0.270$). Moreover, AMC's mean normalized logarithm score only increases
by $9\%$ when $N_T$ is multiplied by 50. It means that this is a very efficient method, thanks to the large-scale structures in the
phase space of the AMOC model (see Fig. 2a): the committor on neighbouring points is consistent and thus averaging it is a
reliable method.

AMC is fairly easy to optimise since it relies on a single parameter: the number of analogues $K$. We tested different values
of $K$, ranging from $K = 5$ to $K = 250$, but we retained only the results for $K = 25$, because they gave the best results, both in
terms of logarithm and difference score. For the largest training dataset, we find a mean normalized logarithm score of $0.708$
for $K = 250$ up to $0.718$ for $K = 25$.

DGA has an even better score than AMC for $N_T \leq 50$, although the relative difference of both scores, of less than $9\%$
is largely within both errorbars. The gap between the 5th and 95th percentile of DGA is also slightly larger than for AMC
(respectively $0.337$ against $0.331$), because it is more skewed towards larger scores. It shows that DGA may tend to produce
sharper transitions in the committor between the on-states and off-states. The similarity between the scores of AMC and DGA
may be explained if we consider that they both use a sampling of the phase space to estimate the committor. The scores are not
shown for $N_T > 150$ because they could not be computed due to the too large computation time (see Sect. 3.1.5).

However, the most important feature of the normalized logarithm score of DGA is that it decreases as $N_T$ increases. It
means that the DGA method becomes less and less efficient as its training set grows larger. We have not found any satisfying
explanation for this seemingly paradoxical behaviour. However, the normalized logarithm score of DGA for $N_T = 10$ is the
second-best score of all, just behind the score of FFNN for $N_T = 500$ ($0.722$ for DGA against $0.729$ for FFNN).

The performance of the FFNN (orange) is exactly the expected one as it is well known that machine learning poorly performs
when trained with too little data. Here, FFNN yields a normalized logarithm score of $0.026$ when trained with the smallest
dataset, which is equivalent to predicting the climatology everywhere. Moreover, the committor estimates are very inconsistent,
heavily depending on the trajectory: the 5th percentile is $-0.862$ and the 95th percentile is $0.516$. But as the size of the training
set increases, the FFNN can extract more complex features from the data. As a result, the score increases fast and the errorbars
shrink. For $N_T \geq 200$, FFNN performs at least as well as AMC and keeps improving although this difference is negligible
compared to errorbars. For instance, for $N_T = 500$, the respective scores of AMC and FFNN are $0.718$ and $0.730$. Their 95th
percentiles of score are similar (respectively $0.871$ and $0.866$) but the distribution of scores of FFNN is overall narrower than
that of AMC, with a 5th percentile of $0.600$ against $0.551$ for AMC.

The other machine learning technique, RC (purple), also has a low score when trained with insufficient data and it increases
with the size of the training set. However, for $N_T = 10$, its normalized logarithm score is already $0.531$, much better in average
than FFNN (above the 95th percentile of its scores). Then for $N_T \geq 30$, the evolution of the average normalized logarithm
score of RC exactly follows that of FFNN, only differing by $0.8\%$ ($N_T = 50$) to $3\%$ ($N_T = 500$), however remaining lower
than the score of FFNN for $N_T \leq 100$. RC also exhibits a narrower gap than FFNN between the 5th and 95th percentiles of its
score, by more than $20\%$ for $30 \leq N_T \leq 150$ and by $7\%$ or less for larger $N_T$. For $N_T \geq 200$, the skill of RC reaches a plateau





at 0.707 and stops improving. This may be due to the limited learning capacity of the vector containing the nonlinear features
(see Appendix C).

The difference score for these methods is shown on Fig. 3b. The scores of the machine learning methods (FFNN and RC) increase overall as the size of training set increases: from $0.767$ ($N_T = 10$) to $0.964$ ($N_T = 500$) for FFNN and from $0.758$ ($N_T = 10$) to $0.954$ ($N_T = 400$) for RC. As was already the case with the normalized logarithm score, the machine learning methods are by far the poorest-performing ones when not trained with enough data. But as $N_T$ increases, FFNN becomes better 430 than the phase-space-sampling-based methods. The score of RC strongly decreases from $N_T = 10$ to $N_T = 20$, from $0.758$ to $0.506$, due to the drop in its 5th percentile from $0.634$ to $0.201$. But for $N_T \geq 50$, the average score differs between FFNN and RC by only $2\%$, a smaller gap than FFNN's errorbars. As was already observed in the normalized logarithm score, RC skill seems to reach a plateau after $N_T = 200$, around a score of $0.953$.

AMC performs clearly better in average than FFNN and RC for $N_T \leq 50$, with a difference score of $0.895$ to $0.931$. Its 435 errorbars, however, overlap those of both machine learning methods. The score of AMC then keeps on steadily increasing with $N_T$ but slower, reaching $0.948$ for $N_T = 200$ to $0.950$ for $N_T = 500$.

Finally, once again, the score of DGA decreases as $N_T$ increases, from $0.914$ for $N_T = 10$ down to $0.904$ for $N_T = 100$. It is thus the best method for $N_T \leq 20$ and the poorest-performing method in average for $N_T \leq 50$. Its average score for $N_T = 100$ is even below the 5th percentile of both AMC and FFNN.

However, the scores ranking between AMC, FFNN and RC might not be so meaningful since the largest difference between their average scores is not as large as the errorbars of FFNN, the narrowest of all.

### 3.1.5 Computation time

Figure 3c shows the training times of each method and Fig. 3d presents the testing times. In practice, both are complementary and equally important. As expected, all methods have a training time scaling with the size of the training set. This is expected 445 because the methods need to process an increasing amount of data when the size of the training set increases. FFNN is about 98 times slower than AMC for the smallest training set. The training in this case takes about $2.44$ seconds instead of $0.025$ seconds for AMC. However, for the largest training set, this difference has shrunk to 5 times only. When $N_T = 500$, FFNN takes about 125 seconds to be trained instead of 25 seconds for AMC. This is because the training time for AMC increases fast for $N_T > 50$, since this method implies computing the largest eigenvectors and eigenvalues of the matrix $\tilde{G}$ (see Sect. 2.4), 450 which size depends on $N_T$. This process appears to scale with the size of the dataset, and not as good as running a simple neural network.

For each value of $N_T$, FFNN has been trained 20 times independently because its training process involves randomness due to the gradient-descent algorithm. However, the 5th and 95th percentiles cannot be seen on the plot because their gap is very small: for $N_T \geq 50$, it is always less than $3.7\%$ of the average score.

As for RC (purple) it is interesting to see how fast the training of this machine learning method is. The training time is the main fallback of the FFNN because of the weights-updating algorithm run after each batch and the large number of weights. In the case of RC, the training mostly amounts to computing the nonlinear features that are extracted from the data at every





time step. This can be achieved in a single NumPy operation for the whole training set, which is thus very well optimized. As a result, RC scales much better with an increasing $N_T$ than FFNN: training RC is 47 times faster than training FFNN for
$N_T = 10$ but it is up to 240 times faster for $N_T = 500$.

DGA is the method which training time has the worst scaling with $N_T$: it increases from 0.30 seconds for $N_T = 10$ to 119 seconds for $N_T = 150$, hence a multiplication by almost 400. It becomes the longest training time for $N_T \leq 100$ and quickly blows up, due to the large matrix operations that this method implies (see Appendix D). On top of that, if we also want the method to optimize $d, \epsilon_0$ and $\epsilon$ (see Appendix D) by itself, then the training time surges to 24.5 seconds for $N_T = 75$, making
it the slowest method of all.

The different methods can be separated into two groups when looking at their testing times (Fig. 3d). DGA has a much larger testing time than AMC, FFNN and RC.

Once again, the testing time of DGA scales with the size of the training set and quickly blows up as well. Indeed, the modes used during the test phase are of the size of the training set, which makes the testing phase increasingly time-expensive. For
$N_T = 10$, this testing time is 0.36 seconds, up to 6.54 seconds for $N_T = 150$. For $N_T = 10$, it is already 27 times as long as the testing time of FFNN, yet the second-longest.

In the second group, for $N_T \leq 200$, FFNN has the largest testing time, with an average of 0.01 seconds. Once again, its errorbars are extremely narrow, showing how consistent this method is, the gap between its 5th and 95th percentile being only 2.3% of the average score. For $N_T > 200$, it is AMC that has the largest testing time of this group because it scales with the
size of the training set (the size of the KD-tree used during the testing phase increases with the size of the training set). Indeed, for $N_T = 10$, the testing time of AMC is 3 times lower than that of FFNN, but up to 1.7 times longer for $N_T = 500$. The testing time of RC is overall the lowest of this group, consistently between $5.1 \times 10^{-3}$ seconds and $5.2 \times 10^{-3}$ seconds, only contained within the errorbars of AMC (yet below its average) for $N_T \leq 50$.

### 3.2  Double-gyre model

#### 3.2.1  Phase space analysis

For $\sigma \in [0.3, 0.48]$, the double-gyre model is in a bistable regime with two fixed points which we refer to as an "up" state and a "down" state. They are respectively defined by $(A_1^{up}, A_2^{up}, A_3^{up}, A_4^{up})$ and $(A_1^{down}, A_2^{down}, A_3^{down}, A_4^{down})$. These states result in two steady states of the system's stream function, which shows "up" and "down" jet states (see Fig. 4a). As we add white noise, the system will undergo transitions between these two states. If we define "up" to be an "on-state" and "down" to be
an "off-state", we have a similar terminology as in the AMOC model. The committor function here is similarly defined as the probability to reach the "off-state" before the "on-state".

Although of lower dimension than the AMOC model, the double-gyre model exhibits a phase space with a more complex structure. Figure 4b displays a snapshot of it for $\sigma = 0.3$ and $\gamma = 0$. The phase space is partitioned into two zones: one where all initial conditions will lead to the on-state and the other where all initial conditions will lead to the off-state. To compute
these zones, we choose a fixed value of $A_3 = 0.069$ and sample $10^4$ random values of $(A_1, A_2, A_4)$ from a uniform distribution



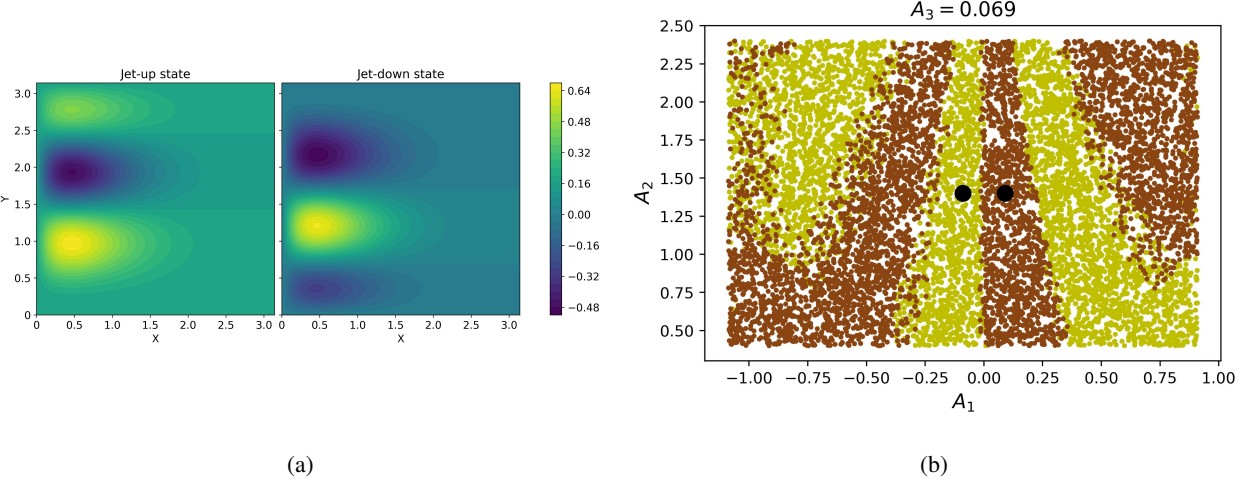

(a)                    (b)

**Figure 4.** (a) Stream functions of the jet-up and jet-down steady states of the double-gyre model, for $\sigma = 0.45$.
(b) Phase space of the double-gyre model ($\sigma = 0.3, \gamma = 0$) in the plane $(A_1, A_2)$ for a fixed value of $A_3$. Black dots correspond to both steady states. The yellow dots are the initial conditions leading to the on-state. The brown dots show the initial conditions leading to the off-state.

on the intervals $[A_1^{up} - 1, A_1^{up} + 1], [A_2^{up} - 1, A_2^{up} + 1]$ and $[A_4^{up} - 1, A_4^{up} + 1]$. Figure 4b is a projection on the plane $(A_1, A_2)$. In this figure, the black dots represent both steady states, the yellow dots represent the initial conditions leading to the on-state and the brown dots the initial conditions leading to the off-state. If we imagine that the structure displayed in this figure evolves along the $A_3$-axis, we see how much more complex this structure is compared to the AMOC model. Overlap between the

yellow and brown dots originates from the projection of the different values of $A_4$ on the $(A_1, A_2)$ plane.

In the following study of this model, all trajectories will be computed with $(\sigma, \gamma) = (0.45, 0.198)$.

### 3.2.2   Train and test datasets

The generation procedure of the train and test datasets used for studying the double-gyre model is similar to that used for the AMOC model. The standard length of all trajectories is 5000 time steps and the test set also consists of 100 such independent

trajectories. The logarithm and difference scores are averaged over these trajectories.

In the case of the double-gyre model, generating the training sets is simpler than for the AMOC model as we do not need to pay attention to different collapsed states. We can just simulate a single very long trajectory containing 500 transitions. All other training sets, containing $N_T = 10, 20, 30, 50, 75, 100, 150, 200, 300, 400$ transitions simply consist of the first $N_T$ transitions of that very long trajectory.





### 3.2.3  Application of the different methods


Once again, we will compare in the following sections the performance of each method against the size of the training set. AMC, DGA and RC were applied on the full training sets. We performed 20-cross validation for the training of FFNN. The parameters of RC and DGA are also optimized by hand and can be found respectively in Table C1 and Table D1.

However, for this model, the implementation of AMC is slightly different than for the AMOC model. As is done in Lucente 510 (2021), the distance used to compute the analogues of every state is normalized by the variance of the distribution of each variable. The choice for this normalization simply comes from the observation that, in this case, unlike the AMOC model, doing so improves the skill of this method.

For the double-gyre model, all methods are applied on the full set of variables: $A_1, A_2, A_3$ and $A_4$.

### 3.2.4  Skill

First of all, we can see that the curves of AMC in all plots of Fig. 5 (blue curves) stop after $N_T = 200$. This is due to the too long computation time of the eigenvalues and eigenvectors of the matrix $\tilde{G}$ (cf. Sect. 2.4) when training this method. Indeed, we observe in the double-gyre model that trajectories spend less time in the on/off-states compared to the AMOC model. As a result, the trajectories of the double-gyre model contain more transition states and $\tilde{G}$ is larger than its equivalent in the AMOC model. For instance, in the latter model, for $N_T = 500$, $\tilde{G}$ is a square matrix of size $13988 \times 13988$. In the double-gyre model, 520  however, for $N_T = 200$, $\tilde{G}$ is already of size $25359 \times 25359$. When considering the normalized logarithm scores for the double-gyre model, shown in Fig. 5a, we see that AMC is no longer one of the best-performing method. It is even the worst-performing method in average for $N_T > 100$. For $N_T \leq 100$, its score increases from $-0.430$ up to $0.032$. It is thus only for $N_T \geq 100$ that AMC can perform at least as well as the climatology. Moreover, its maximum normalized logarithm score is only $0.167$ for $N_T = 200$. Furthermore, the distribution of scores of AMC exhibits a huge spread: the difference between its 95th percentile 525  and its 5th percentile is $1.702$ for $N_T = 20$, down to $0.746$ for $N_T = 200$. For such a large dataset, it corresponds to the largest spread of all methods. Moreover, the optimal value of $K$ here is $K = 15$, instead of $K = 25$ for the AMOC model. When increasing the value of $K$, the normalized logarithm score dramatically decreases: it is in average $-0.012$ for $K = 20$, $-0.083$ for $K = 25$ and keeps decreasing. In other words, for $K \geq 20$, AMC performs worse than the climatology. This may be explained by the more complex structure of the phase space of the double-gyre model (see Fig. 4b).

Here, FFNN performs even more poorly as in the AMOC model when trained with too little data (score of $-0.690$ for $N_T = 10$). FFNN is even the poorest-performing method up to $N_T = 100$ and only manages to give a better estimate of the committor than AMC for $N_T \geq 150$. Its maximum score, for $N_T = 500$, is $0.360$. Moreover, the gap between the 5th and 95th percentiles of this score is huge for $N_T \leq 75$ (more than $1.89$, systematically larger than that of AMC) but fastly shrinks down to $0.6$ in average for $N_T \geq 150$. From $N_T = 75$ to $N_T = 150$, the average normalized logarithm score of FFNN also increases 535  from $-0.218$ to $0.297$, and becomes the second-best method (while it is the worst for $N_T \leq 75$), this value of $N_T$ acting as a sort of threshold above which the FFNN is much more efficient.

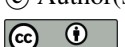


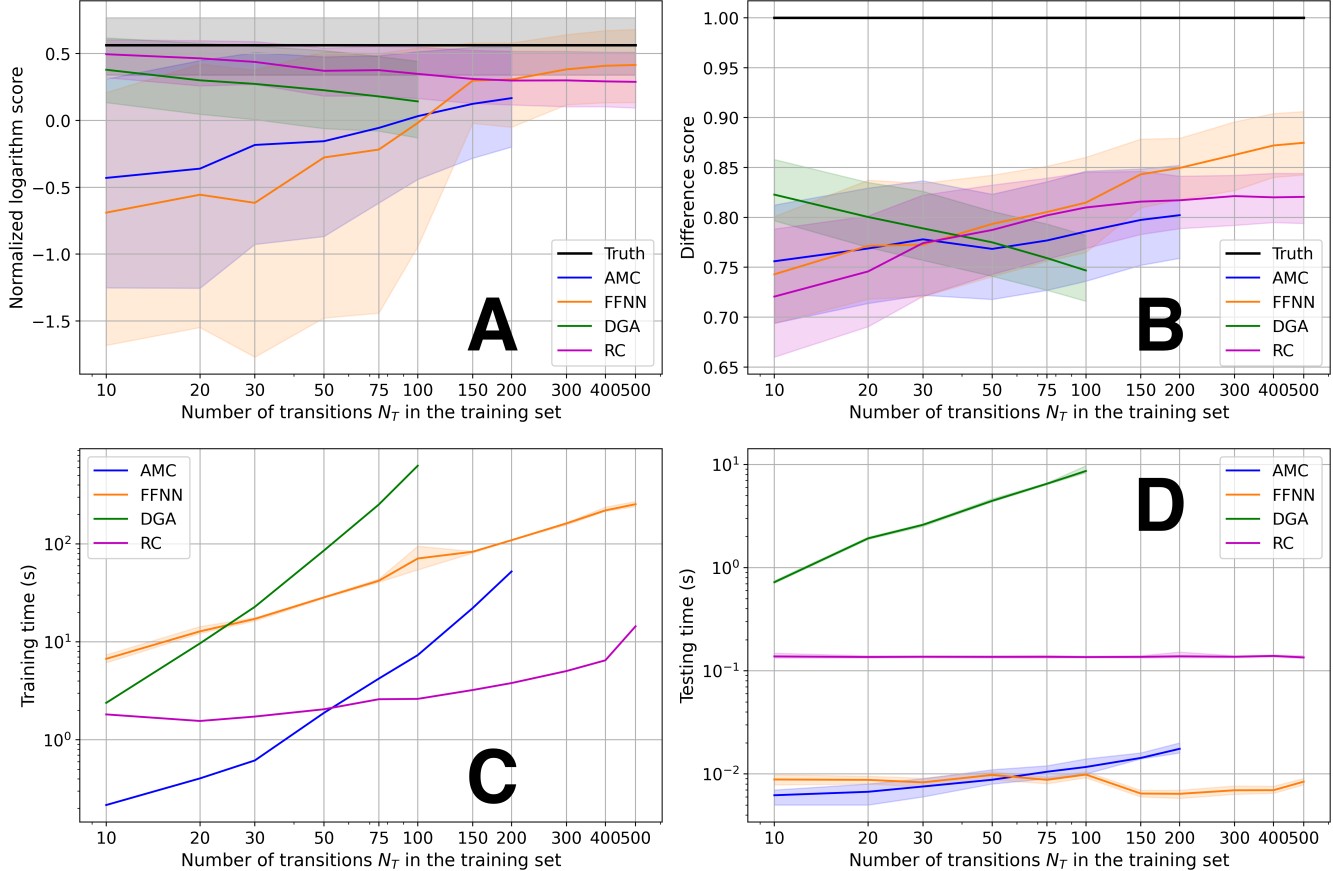

**Figure 5.** Comparison of the performance and computation time of all four methods on the double-gyre model.

(a) The normalized logarithm scores of each method.

(b) The difference scores of each method.

(c) The training times of each method.

(d) The testing times of each method.

In (a), (b) and (d), each curve is the average over the score of the 100 test trajectories. The shaded areas around each curve are their 90% most probable values, between the 5th and 95th percentiles. In (c), the shaded area is only provided for FFNN, since all other methods are only trained once.

Once again, the curve of DGA (shown in green) stops at $N_T = 100$ due to the too large matrix computation. Once again, its normalized logarithm score decreases as $N_T$ increases, from $0.378$ for $N_T = 10$ down to $0.142$ for $N_T = 100$. However, on that interval, DGA remains the second-best method after RC.

RC also has a decreasing normalized logarithm score as $N_T$ increases, from $0.495$ for $N_T = 10$ down to $0.288$ for $N_T = 500$, but its difference score increases in the meantime, which we explain at the end of this section. For $N_T = 10$ to $N_T = 500$, the gap between its 5th and 95th percentile also increases from $0.283$ to $0.416$. This method thus shows the opposite behaviour





compared to the other methods involving training. We will provide an interpretation of this phenomena when looking at the difference score of that method.

The difference score of each method (Fig. 5b) is further away from the perfect estimate than their equivalent in the AMOC model, once again showing that the committor estimation task in this model is more difficult.

DGA is only the best-performing method for $N_T \leq 30$, although its difference score drops from $0.823$ to $0.789$. The difference score of AMC only increases from $0.756$ in average to $0.802$ between $N_T = 10$ and $N_T = 200$, making it the worst-performing method for $N_T \geq 150$. For $N_T \geq 50$, FFNN is on average the best-performing method, with a score of only $0.875$

at its maximum, for $N_T = 500$. This maximum score is much further away from the score of the perfect estimate ($1.0$) than in the AMOC model (maximum score of $0.964$), showing once again that computing the committor function in a more complex phase space is a much more difficult task. However, FFNN is clearly the best-performing method when trained with a large training set ($N_T \geq 400$) since there is barely any overlap between its errorbar and that of RC, the second-best performing method.

For $N_T \leq 30$, RC is the worst-performing method with a score of less than $0.774$ but it closely follows the difference score of FFNN until $N_T = 100$. RC then reaches a performance plateau at $0.820$ with the narrowest errorbars of all methods for $N_T \geq 150$ (having a width between $0.05$ and $0.06$). So, as $N_T$ increases, the difference score of RC increases and its errorbars shrink but its normalized logarithm score decreases. This may be explained by the fact that trajectories in this model show a lot of "aborted transitions": for instance, starting from the on-state, the systems evolves towards the off-state but the noise drives it

back to the on-state. In the committor function, starting from $q(x(t_0)) = 0$, the value of the committor increases, reaches a peak $q(x(t_1)) < 1$ for $t_1 > t_0$ and then goes back to 0. This is the reason why the normalized logarithm score of these trajectories is lower than that of the trajectories in the AMOC model: such "false transitions" are heavily penalized by this score. It also explains the behaviour of RC: as $N_T$ increases, RC increasingly tries to mimic this kind of features, which is penalized by the normalized logarithm score, although its committor estimate overall better matches the true committor.

**3.2.5 Computation time**

The computation times for the double-gyre model are shown in Fig. 5c and Fig. 5d. As was already the case for the AMOC model, the training time of RC scales better with the size of the training set than that of FFNN (Fig. 5c): FFNN takes $2.7$ times longer to be trained for $N_T = 10$ and is over 16 times slower than RC for $N_T = 500$. RC only takes $1.82$ seconds to be trained for $N_T = 10$, up to $14.42$ seconds for $N_T = 500$.

During the training of AMC, due to the fast growing size of the matrix $\tilde{G}$ as $N_T$ increases, this method has again a worse scaling with the size of the training set than FFNN. Its training time is multiplied by 242 from $N_T = 10$ ($0.22$ seconds) to $N_T = 200$ ($52.26$ seconds). As a comparison, over that same range, the training time of FFNN is only multiplied by 38 and that of RC by $8$.

For $N_T \leq 20$, DGA is faster to train than FFNN, with a training time of less than 10 s. However, for $N_T \geq 30$, it becomes

the slowest method of all by far, its training time ranging from 23 seconds to 627 seconds for $N_T = 100$. Between $N_T = 10$ and $N_T = 100$, this training time is multiplied by 263 so its scaling with the amount of data is even worse than for AMC.





The testing times are shown in Fig. 5d. The testing time of RC is $0.136$ seconds in average, which is a significant increase (26 times larger) compared to the AMOC model. It is largely due to the increase in dimension of the features vector (from 330 to 1716 coefficients). The testing time of AMC is less affected by this change of model, since it only consists in a search in a KD-Tree. For $N_T \leq 50$, AMC has the shortest testing time, from $0.006$ seconds for $N_T = 10$ up to $0.009$ seconds for $N_T = 50$.

As was already the case in the AMOC model, FFNN also has a very short testing time of about $0.008$ seconds. The testing has even decreased by $38\%$ compared to the AMOC model, making FFNN the method with the shortest testing time for $N_T \geq 75$. Finally, the testing time of DGA ranges from $0.72$ seconds for $N_T = 10$ to $8.64$ seconds for $N_T = 100$, blowing up again afterwards. This is 5 to 64 times longer than the RC, which is the second longest. Its maximum testing time is only $2.1$ seconds longer than for the AMOC model (thus increasing by $32\%$), although the number of modes involved has been multiplied by 20. It shows that the number of modes only has a limited impact on the computation time, which mainly depends on the size of the training set.

## 4   Summary and Discussion

The present work intends to evaluate and compare several existing methods to estimate the committor function from trajectory data. Having a good estimate of the committor function is crucial in order to ensure maximum efficiency and accuracy of a rare-event algorithm such as TAMS. Using these kind of algorithms is a very promising solution to the computation of probabilities of rare transitions in complex dynamical systems, such as a potential collapse of the AMOC in high-dimensional ocean-climate models. We compared the Analogue method (AMC) with a simple feedforward neural network (FFNN), a Reservoir Computing (RC) method and a Dynamical Galerkin Approximation (DGA) scheme. Two models, an AMOC box model and a double-gyre model were used for their evaluation, where the phase space dynamics of the double-gyre model is more complex than that of the AMOC model.

Although efficient in the AMOC model, AMC is very slow and not so robust in more complex settings such as the double-gyre model. The sampling of the phase space indeed becomes difficult when it displays complex structures. This result may be related to what Lucente et al. (2022a) observed in the Jin-Timmermann model: there are certain zones of the phase space where the committor function displays a complex, fine scale structure, which we cannot expect AMC to predict accurately due to its analogues appproximation. Even the testing phase, that uses a search through a KD-tree quickly becomes very computationally expensive with dimensionality and requires a lot of training data.

FFNN proves a very robust method, that can adapt to complex phase spaces. Its main drawback is the time it takes to be trained and the amount of data needed to obtain an adequate estimate of the committor. However, once trained, it is a very fast method, that also provides the best estimate of the committor. The RC method is the most naive of all, extracting nonlinear features from a trajectory and performing a linear regression on them. This method is strikingly efficient, considering how simple it is. When well optimized, its results may compete with those of the FFNN but it is much faster to train. However, it has a limited learning capacity and reaches a performance plateau which makes a difference with FFNN when trained with a lot of data.



The DGA method shows a strange behaviour that we could not explain: its performance decreases as the size of the training set increases. However, for the lowest value of $N_T$ tested, DGA is one of the best-performing methods, competing with the machine learning methods trained with $N_T = 500$. This method is thus efficient in terms of number of transitions: it requires a limited amount of data to compete with the best-performing methods. Its main limit is its computing time and that its parameters need to be tweaked by hand to be fully optimized.

We compared these methods using two scores: the normalized logarithm score and the difference score. Although the latter is easier to interpret, it will in general never be computable because it requires to know the true committor. For more complex models, we will thus have to rely on the normalized logarithm score. We found that they do not provide the exact same information: in particular, they rank differently the methods. However, in general, the improvement in the skill of most methods can be read accordingly in both scores. We only found one exception: RC in the case of the double-gyre model, where the

normalized logarithm score wrongly indicates a loss in performance as $N_T$ increases.

By applying rare event algorithms to more sophisticated, high-dimensional models, it is likely that long (and expensive) simulations contain few, or even no transitions. This is a major problem because AMC, FFNN and DGA all rely on reactive trajectories to be trained and then estimate the committor. If there are no transitions in the data, the neural network only sees one class of events so it can never predict a transition. In addition, it can be easily demonstrated that AMC and DGA fail as

well. So, we may need extra-long costly simulations to be able to apply these methods, all the more so as AMC and FFNN require a sufficient number of transitions to be trained properly.

In this setting, DGA and RC are promising. Indeed, although DGA needs transitions to be trained, we showed that much less data is required than for any other method in order to obtain good results. Moreover, we do not necessarily need to see complete transitions in the trajectories. Only certain relevant areas in the phase space need to be explored and sampled

(although determining which ones precisely is not obvious in general), which can be done by stacking shorter trajectories that do not necessarily transition from one state to another. This approach is the one developed in Finkel et al. (2021). It consists in first simulating a very long control trajectory and then drawing $N$ samples from it. These samples will serve as initial conditions for short trajectories on which DGA is then applied. Finkel et al. (2021) show very good agreement between this approach and a Monte-Carlo approach, all the more so as their approach allows estimating the committor on many states at the same time and

can be parallelized. RC may also be interesting because at its core lies a simple linear regression. It is thus the only method that makes no assumption on transitions: the training committor will be fitted whatever the value of the probabilities. The problem, on the other hand, is that we need to compute the committor for training, which is what we are trying to avoid because of its cost. It might then be interesting to use a combined approach: for instance, compute a first estimate of the committor using DGA and use it to train RC.

The next step of this work would be to combine these data-driven estimates of the committor function with TAMS to actually compute rare-event probabilities. However, it requires some extension of the present study. We already mentioned the problem of regimes where long trajectories contain only few to no transitions. Moreover, we may want to compute transition probabilities for different parameters of the model, as is done by Castellana et al. (2019) for the AMOC box model or by Baars et al. (2021) with the model from den Toom et al. (2011). In this case, the dynamics of the model change with each set





of parameters and we have to take it into account during the training of the method we will be using. Such an adaptation to changing dynamics has recently been implemented for RC by Kong et al. (2021) but we can also think of other approaches. For instance, FFNN can be combined with transfer learning: it consists in first training it in a regime where we have a lot of data and the estimation is easy. Then, we use that learnt knowledge to (warm) start the training in a regime where transitions are much less probable. Jacques-Dumas et al. (2022) have shown that this method at least reduces the computation time for the

prediction of extreme events.

      A related approach consists in building a feedback loop between a rare-event algorithm and a data-driven committor function estimation method. The estimate of the committor yielded by the latter is used by the former to generate more data in order to improve the committor estimate. This idea has already been implemented by Nemoto et al. (2016) and more recently by Lucente et al. (2022b). The latter in particular have coupled AMC with AMS, testing for the first time this approach in a

model with complex dynamics. Another interesting work is that of Du (2020), where AMS was coupled to Mondrian Forests, a relatively new type of random forests method (Lakshminarayanan et al., 2014) that can also be used to estimate the committor function. The advantage of Mondrian Forests is that this allows to apply online learning: new data samples can be provided one after the other to incrementally improve the estimation of the committor and the order in which they are provided does not matter. The power of this property is clear when it comes to coupling with TAMS: the committor estimation can be improved

every time a new clone trajectory is simulated. However, all these coupling approaches have until now only been applied to low-dimensional systems and might prove computationally expensive in the case of high-dimensional models.

      Another extension of this work would be to consider non-autonomous dynamics. Once again, this extension can be achieved from several viewpoints. Firstly, if the objective is to compute the probability that the AMOC collapses within a certain time frame, Lucente (2021) proposes to add a time dimension to the system's phase space. Suppose the goal is to compute the

probability to reach a set $A$ of the phase space before another set $B$ and before a time $T_{max}$. Either a time-independent committor can be used in combination with TAMS or the problem can be reformulated as: the probability to reach a set $A$ of the newly-expanded phase space before another set $\tilde{B} = B \cup \{t \mid t \geq T_{max}\}$. This new formulation overrides of course the use of TAMS but may pose additional issues concerning the training data. The second viewpoint is more general: it consists in studying the committor in a non-autonomous system where either the equations explicitly depend on time or the sets $A$ and

$B$ themselves depend on time. Helfmann et al. (2020) and Sikorski et al. (2021) propose frameworks to work with such time-dependent committor functions. Such a generalization is especially of interest when working on climate problems involving global warming.

      The long-term objective of such a study and extensions would be to apply TAMS and committor function estimation to much more complex models, such as Earth system Models of Intermediate Complexity (EMIC) or even General Circulation Models

(GCM). Their complexity are nowhere comparable to the models featured in this work, with a dimensionality of the order of at least $10^6$. By looking at partial differential equations, or even intermediate complexity climate models, Bouchet et al. (2019) and Ragone et al. (2018) have already applied TAMS on such high-dimensional systems. However, thanks to a simpler phenomenology, they could design suitable score functions and TAMS has never been used in combination with committor functions in these cases. Extending the methods presented here will be a real challenge, regarding for instance optimization and





the limited amount of data available. It is, however, an interesting scientific perspective to gain more insight into these models'
dynamics through the probability of occurrence of rare events.

## Appendix A: AMOC box model

The equations of the AMOC model are:

$$
\begin{aligned}
\frac{\mathrm{d}(V_t S_t)}{\mathrm{d}t} = {} & q_s(\theta(q_s)S_{ts} + \theta(-q_s)S_t) + q_u S_d \\
& - \theta(q_n)q_n S_t + r_s(S_{ts} - S_t) \\
& + r_n(S_n - S_t) + 2E_s S_0
\end{aligned}
\tag{A1}
$$


$$
\begin{aligned}
\frac{\mathrm{d}(V_{t_s} S_{ts})}{\mathrm{d}t} = {} & q_{Ek}S_s - q_e S_{ts} - q_s(\theta(q_s)S_{ts} + \theta(-q_s)S_t) \\
& + r_s(S_t - S_{ts})
\end{aligned}
\tag{A2}
$$

$$
\begin{aligned}
\frac{\mathrm{d}(V_n S_n)}{\mathrm{d}t} = {} & \theta(q_n)q_n(S_t - S_n) + r_n(S_t - S_n) \\
& - (E_s + E_a)S_0
\end{aligned}
\tag{A3}
$$

$$
\begin{aligned}
\frac{\mathrm{d}(V_s S_s)}{\mathrm{d}t} = {} & q_s(\theta(q_s)S_d + \theta(-q_s)S_s) + q_e S_{ts} \\
& - q_{Ek}S_s - (E_s - E_a)S_0
\end{aligned}
\tag{A4}
$$


$$
(A + \frac{L_{x_A}L_y}{2})\frac{\mathrm{d}D}{\mathrm{d}t} = q_u + q_{Ek} - q_e - \theta(q_n)q_n
\tag{A5}
$$

$$
S_0 V_0 = V_n S_n + V_d S_d + V_t S_t + V_{ts}S_{ts} + V_s S_s
\tag{A6}
$$



The function $\theta(x)$ is here defined as the Heaviside function, equal to 1 if $x > 0$ and 0 otherwise. The flows between the boxes are defined as:

$$q_{Ek} = \frac{\tau L_{xS}}{\rho_0 |f_S|} \tag{A7}$$

$$q_e = A_{GM} \frac{L_{xA}}{L_y} D \tag{A8}$$

$$q_s = q_{Ek} - q_e \tag{A9}$$

$$q_n = \eta \frac{\rho_n - \rho_{ts}}{\rho_0} D^2 \tag{A10}$$

$$q_u = \frac{\kappa A}{D} \tag{A11}$$

where the density of the box $i$ is defined as:

$$\rho_i = \rho_0 (1 - \alpha(T_i - T_0) + \beta(S_i - S_0)) \tag{A12}$$

The volumes of the boxes $t$, $ts$ and $d$ are defined as:

$$V_t = AD \tag{A13}$$

$$V_{ts} = \frac{L_{xA} L_y}{D} \tag{A14}$$

$$V_d = V_0 - V_t - V_{ts} - V_n - V_s \tag{A15}$$

Finally, we present in Table A1 the constants and values of the parameters used here.

**Appendix B: Double-gyre model**

The stream function of the double-gyre is written as

$$\Psi(x, y, t) = \sum_{k=1}^{4} A_k(t) \Phi_k(x, y) \tag{B1}$$

where the coefficients $A_k(t)$ are computed from

$$\int_0^\pi \int_0^\pi [\text{Eq. (2)}] \Phi_k \, \mathrm{d}x \mathrm{d}y \tag{B2}$$

As a result, the mode amplitudes $A_k$ obey the following set of ODEs:





**Table A1.** Reference constants and parameters of the AMOC model.

|  |  | Parameters used in the model |
| --- | --- | --- |
| $V_0$ | $3 \times 10^{17}$ m$^3$ | Total volume of the basin |
| $V_n$ | $3 \times 10^{15}$ m$^3$ | Volume of the Northern box |
| $V_s$ | $9 \times 10^{15}$ m$^3$ | Volume of the Southern box |
| $A$ | $1 \times 10^{14}$ m$^2$ | Horizontal area of the Atlantic pycnocline |
| $L_{xA}$ | $1 \times 10^7$ m | Zonal extent of the Atlantic Ocean at its Southern end |
| $L_y$ | $1 \times 10^6$ m | Meridional extent of the frontal region of the Southern Ocean |
| $L_{xS}$ | $3 \times 10^7$ m | Zonal extent of the Southern Ocean |
| $\tau$ | 0.1 N m$^{-2}$ | Average zonal wind stress amplitude |
| $A_{GM}$ | 1700 m$^2$ s$^{-1}$ | Eddy diffusivity |
| $f_S$ | $-10^{-4}$ s$^{-1}$ | Coriolis parameter |
| $\rho_0$ | 1027.5 kg m$^{-3}$ | Reference density |
| $\kappa$ | $10^{-5}$ m$^2$ s$^{-1}$ | Vertical diffusivity |
| $S_0$ | 35 psu | Reference salinity |
| $T_0$ | 5 K | Reference temperature |
| $T_n$ | 5 K | Temperature of the Northern box |
| $T_{ts}$ | 10 K | Temperature of the box $ts$ |
| $\eta$ | $3 \times 10^4$ m s$^{-1}$ | Hydraulic constant |
| $\alpha$ | $2 \times 10^{-4}$ K$^{-1}$ | Thermal expansion coefficient |
| $\beta$ | $8 \times 10^{-4}$ psu$^{-1}$ | Haline contraction coefficient |
| $r_s$ | $1 \times 10^7$ m$^3$ s$^{-1}$ | Transport by the Southern subtropical gyre |
| $r_n$ | $5 \times 10^6$ m$^3$ s$^{-1}$ | Transport by the Northern subtropical gyre |
| $E_s$ | $0.17 \times 10^6$ m$^3$ s$^{-1}$ | Symmetric freshwater flux |

$$\frac{\mathrm{d}A_1}{\mathrm{d}t} = c_1 A_1 A_2 + c_2 A_2 A_3 + c_3 A_3 A_4 - \mu_1 A_1 \tag{B3}$$

$$\frac{\mathrm{d}A_2}{\mathrm{d}t} = c_4 A_2 A_4 + c_5 A_1 A_3 - c_1 A_1^2 - \mu_2 A_2 + c_6 \sigma (1 + \gamma \zeta) \tag{B4}$$

$$\frac{\mathrm{d}A_3}{\mathrm{d}t} = c_7 A_1 A_4 - (c_2 + c_5) A_1 A_2 - \mu_3 A_3 \tag{B5}$$

$$\frac{\mathrm{d}A_4}{\mathrm{d}t} = -c_4 A_2^2 - (c_3 + c_7) A_1 A_3 - \mu_4 A_4 \tag{B6}$$

where white noise $\zeta(t)$ is added and the fixed parameters are shown in Table B1.




**Table B1.** Fixed parameters of the double-gyre model.

| $c_1$ | $c_2$ | $c_3$ | $c_4$ | $c_5$ | $c_6$ | $c_7$ |
|---|---|---|---|---|---|---|
| 0.020736 | 0.018337 | 0.015617 | 0.031977 | 0.036673 | 0.314802 | 0.046850 |

| $\mu_1$ | $\mu_2$ | $\mu_3$ | $\mu_4$ |
|---|---|---|---|
| 0.0128616 | 0.0211107 | 0.0318615 | 0.0427787 |

**Table C1.** Parameters of the New-Generation Reservoir Network for both models.

| Model | $k$ | $s$ | $p$ | $\alpha$ |
|---|---|---|---|---|
| AMOC | 2 | 1 | 4 | $10^{-9}$ |
| Double-gyre | 2 | 1 | 6 | $10^{-6}$ |

**Appendix C: Nonlinear feature vector in RC**

In this appendix, we explain how Gauthier et al. (2021) compute the feature vector $\boldsymbol{X}$ in the RC approach. $\boldsymbol{X}$ consists in three parts: $c$ is a constant always taken to 1, $\boldsymbol{X}_{lin}$ is the linear part of the feature vector and $\boldsymbol{X}_{nonlin}$ is its nonlinear part. They are concatenated ($\bigoplus$ operation) so that:

$$\boldsymbol{X} = c \bigoplus \boldsymbol{X}_{lin} \bigoplus \boldsymbol{X}_{nonlin} \tag{C1}$$

Firstly, $\boldsymbol{X}_{lin}$ is a concatenation of the current time step and $k$ previous time steps, with a stride of $s$. Let the input trajectory $\boldsymbol{U}$ at the time step $t$ be of the form $\boldsymbol{U}_t = [u_i \ \forall i \in \{1,...,M\}]$. $\boldsymbol{X}_{lin}$ at the time step $t$ is then defined by:

$$\boldsymbol{X}_{lin,t} = \boldsymbol{U}_t \bigoplus \boldsymbol{U}_{t-1\times s} \bigoplus ... \bigoplus \boldsymbol{U}_{t-(k-1)\times s} \tag{C2}$$

Since $\boldsymbol{U} \in \mathbb{R}^{M\times T}$, then $\boldsymbol{X}_{lin} \in \mathbb{R}^{Mk\times(T-ks)}$. The first $ks$ timesteps of the trajectory are the warm-up period, needed to create the first point of $\boldsymbol{X}_{lin}$.

Secondly, $\boldsymbol{X}_{nonlin}$ is defined as a nonlinear function of $\boldsymbol{X}_{lin}$. Gauthier et al. (2021), set $\boldsymbol{X}_{nonlin}$ to contain all unique monomials that can be obtained from the outer product $\boldsymbol{X}_{lin} \bigotimes \boldsymbol{X}_{lin}$. For instance, if $\boldsymbol{X}_{lin,t} = [x,y,z]$, then $\boldsymbol{X}_{nonlin,t} = [x^2, xy, xz, y^2, yz, z^2]$.

Gauthier et al. (2021) then generalize this definition of $\boldsymbol{X}_{nonlin,t}$. A new parameter, $p$, is introduced, corresponding to the maximum degree of the monomials in the nonlinear vector. $\boldsymbol{X}_{nonlin}$ is now defined as:

$$\boldsymbol{X}_{nonlin} = \boldsymbol{X}_{lin} \bigotimes \boldsymbol{X}_{lin} \bigotimes ... \bigotimes \boldsymbol{X}_{lin} \tag{C3}$$

where $\boldsymbol{X}_{lin}$ appears $p$ times.

In the general case, the shape of $\boldsymbol{X}_{nonlin}$ is thus given by the binomial factor $B = \binom{Mk+p-1}{p}$. As a result, $\boldsymbol{X}_{nonlin} \in \mathbb{R}^{B\times(T-ks)}$ and then $\boldsymbol{X} \in \mathbb{R}^{(1+Mk+B)\times(T-ks)}$.





## Appendix D: Basis functions in the DGA

The basis functions are computed from a transition matrix, itself computed from a reactive trajectory:

$$\mathbf{P}_{mn} = \frac{\mathbf{K}_\epsilon(x_m, x_n)}{\sum_n \mathbf{K}_\epsilon(x_m, x_n)} \tag{D1}$$

where $\mathbf{K}_\epsilon$ is a kernel exponentially decreasing as $x_m$ and $x_n$ move further away at a rate depending on $\epsilon$. The submatrix of $\mathbf{P}$ where $x_m$ and $x_n$ belong to $(A \cup B)^c$ is then extracted and its $M$ eigenvectors $\varphi_i$ with the largest eigenvalue are computed. The basis functions $\phi_i(x)$, $i \in [1, M]$ are then defined as:

$$\phi_i(x) = \begin{cases} \varphi_i(x) \text{ for } x \in (A \cup B)^c \\ 0 \text{ otherwise} \end{cases} \tag{D2}$$

Thiede et al. (2019) also provides a method to compute the kernel $K_\epsilon(x_m, x_n)$. We will simply show the equations ; more details can be found in (Berry and Harlim, 2014; Berry et al., 2015; Thiede et al., 2019). The kernel itself is defined by:

$$\mathbf{K}_\epsilon(x_m, x_n) = \exp\left(\frac{-||x_m - x_n||^2}{\epsilon k(x_m)^{-1/d} k(x_n)^{-1/d}}\right) \tag{D3}$$

where

$$k(x_m) = \frac{(2\pi\epsilon_0)^{-d/2}}{N\zeta_0(x_m)^d} \sum_{n=1}^{N} K_0(x_m, x_m, \epsilon_0) \tag{D4}$$

$$K_0(x_m, x_n, \epsilon_0) = \exp\left(\frac{-||x_m - x_n||^2}{2\epsilon_0 \zeta_0(x_m)\zeta_0(x_n)}\right) \tag{D5}$$

$$\zeta_0(x_m) = \frac{1}{k_0} \sum_{l=1}^{k_0} ||x_m - x_{I(m,l)}||^2 \tag{D6}$$

$d, \epsilon_0$ and $\epsilon$ are parameters that we optimized by hand (see Table D1) and $x_{I(m,l)}$ refers to the $l$-th nearest neighbour of $x_m$. Following (Berry and Harlim, 2014; Thiede et al., 2019), we also found $k_0 = 7$ to be a suitable value.

Finally, Thiede et al. (2019) provides a fast method to extend the modes computed on a training set. Suppose the kernel $\mathbf{K}_\epsilon(x_m, x_n)$ has been computed for every point $x_m, x_n$ of the training set. It allowed to compute as well every mode $\phi_i$ (associated to an eigenvalue $\lambda_i$) on each of these points. To extend the mode $i$ on a new point $y$ of the phase space, the following formula can be used:

$$\phi_i(y) = \frac{1}{\lambda_i} \frac{\sum_m \mathbf{K}_\epsilon(x_m, y)\phi_i(x_m)}{\sum_m \mathbf{K}_\epsilon(x_m, y)} \tag{D7}$$

*Code availability.* The Python implementation of both models, all methods and the code producing the result plots can be found at the following adress: https://doi.org/10.5281/zenodo.7380724





**Table D1.** Parameters of the Dynamical Galerkin Approximation for both models.

| Model | $N$ | $d$ | $\epsilon_0$ | $\epsilon$ |
|---|---|---|---|---|
| Cimatoribus | 10 | 0.45 | $2^{15}$ | $2^{-15}$ |
| Double-gyre | 200 | 0.75 | $2^5$ | $2^{-5}$ |

*Author contributions.* All authors conceived the study. VJD carried out the computations, generated all figures, and wrote the first draft of the paper. All authors contributed to the final paper.

*Competing interests.* The authors declare that they have no conflict of interest.

*Acknowledgements.* This project has received funding from the European Union's Horizon 2020 research and innovation programme under the Marie Sklodowska-Curie Grant Agreement No. 956170. The work of F. Bouchet was supported by the ANR grant SAMPRACE, project ANR-20-CE01-0008-01.



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
