# Peer review of "Data-driven methods to estimate the committor function in conceptual ocean models"

_EGUsphere, 2022_

## Author Comment (AC1)

**MS-No.:** egusphere-2022-1362

**Version:** Revision

**Title:** Data-driven methods to estimate the committor function in conceptual ocean models

**Author(s):** Valérian Jacques-Dumas and co-authors

**Point-by-point reply to reviewer #1**

March 20, 2023

We thank the reviewer for their careful reading and for the useful comments and will adapt the manuscript accordingly. Below is a point-by-point reply with the referee's comments in bold font, our reply in italic font and the changes in manuscript in normal font.

1. **- p10: Equation (21) is unclear to me: u and v are time-series. The inner product $<,>$ is, by context, to be interpreted over the state-space and not time? If so, the index i on the rhs is the temporal index. Why is there a sum over the time index? Is this a time-integral? If so, the rhs is a vector and the lhs a scalar? Fortunately, it seems that equation (21) is not used anywhere later on, but arguably the notation in the section could be clarified.**

   *We agree that the notation was confusing and we changed it (lines 261-267). Here, u and v are functions from the state-space to $\mathbb{R}$ and the inner product $<,>$ is defined as an integral over state-space. Equation (21) corresponds to a Monte-Carlo estimate of this inner product using the available time series.*

   We will add more detail about this inner product and the steps that lead to Equation (21) on page 10.

2. **- p22: The explanation around lines 555 to 565 about 'aborted transitions' is not very satisfactory. In particular, the amount of 'aborted trajectories' is quantified by the committor itself by definition. There cannot be a large fraction of trajectories that reach q=0.5 and then abort. Instead, the fraction of trajectories reaching q=0.5 but not transitioning is exactly 0.5, and the same is true for any other value of the committor (for example, 10% of all trajectories reaching q=0.9 eventually abort). It is therefore unclear how one model can show more aborted transitions than another, or what an aborted transition even is. Maybe I am misunderstanding the intention of this paragraph.**

   *We thank the reviewer about this point. Considering the definition of the committor, "aborted transition" was indeed poor phrasing. The intention of this paragraph was to explain why the logarithm score is decreasing for RC as $N_T$ increases while the difference score increases, and to show that the trend of the logarithm score may be misleading when comparing two committor estimates. We also attempted to explain why one score decreases while the other increases. In the double-gyre model, the noise has a larger effect than in the AMOC box model and trajectories take longer to reach either an on or off-state. This causes the average logarithm score of the Monte-Carlo estimate of the committor to decrease in this model, as the trajectory explores larger areas of the phase space and the committor oscillates before reaching 0 or 1.*

   On page 23, we will rewrite this paragraph, remove the expression "aborted transition" and provide a better explanation of the difference between both scores.

---

## Author Comment (AC2)

**MS-No.:** egusphere-2022-1362

**Version:** Revision

**Title:** Data-driven methods to estimate the committor function in conceptual ocean models

**Author(s):** Valérian Jacques-Dumas and co-authors

**Point-by-point reply to reviewer #2**

March 20, 2023

We thank the reviewer for their careful reading and for the useful comments and will adapt the manuscript accordingly. Below is a point-by-point reply with the referee's comments in bold font, our reply in italic font and the changes in manuscript in normal font.

1. **Figure 3: as described in the caption, there should be "confidence intervals" for FFNN training time in subplot (c). Or is it just too narrow to see?**

   *The confidence intervals are plotted the same way on every subplot of Figure 3 and Figure 5. For FFNN in subplot (c), the confidence interval is plotted but indeed too narrow to see. The difference between the $5^{\text{th}}$ and $95^{\text{th}}$ percentiles is less than $8.5\%$ of the average training time for $N_T \leq 30$ and less than $3.7\%$ of the average training time for $N_T \geq 50$. It can only be seen by zooming enough for $N_T = 10$.*

   We will add a sidenote in the caption of Figure 3 explaining that the confidence interval for FFNN is too narrow to see.

2. **It would be helpful to provide more details about the scaling issue in the AMG method. Overall AMG method seems to be rather competitive especially when the number of available data is limited (which is typically the case in real application). Does the computational bottleneck come from the increasingly expensive KD-tree search or the eigenvalue algorithms for the large analogue matrix? For the latter case, one can exploit the sparsity structure in the analogue matrix and use iterative eigen-solvers. Matrices of size 1e4 by 1e4 seem still efficient to work with.**

   *In the case of AMC, the main computational bottleneck stems from the eigenvector algorithm. This step is already implemented using a SciPy sparse matrix function which is a wrapper to the ARPACK function. The latter is an efficient implementation of the Implicitly Restarted Arnoldi Method so AMC already uses an iterative solver and takes advantage of the sparsity of $\tilde{G}$. This solver has, however, a complexity of $O(N^2)$ (if $N$ is the number of samples in the training set) while building and searching through the Kd-tree does not have a larger complexity than $O(N \log N)$.*

   On page 17, we will provide additional details about the time complexity of AMC: "For AMC, building the Kd-tree is at worst $O(N \log N)$ while querying it is about $O(N)$ and the eigenvector search is $O(N^2)$. Thus when $N$ grows large, the training time of AMC mainly depends on the eigenvector search in $\tilde{G}$, although we are using the Implicitly Restarted Arnoldi Method, which already takes advantage of the sparsity of $\tilde{G}$."

3. **The authors focus on the scaling issue from one perspective– the data dimension. However when applying the algorithm to other models, the variable dimension typically plays a critical role and efficient algorithms with moderate complexity growth with the model dimension is indeed the research target in the committor function community. I don't think it is necessary to provide numerical evidence in this angle but I would suggest the author include a brief discussion and clarify this different definition about "high-dimensional" problems.**

   *We thank the reviewer for pointing this out, it is indeed an important distinction to make because the methods don't scale the same way whether we are looking at the data dimension or the variable dimension. For AMC, the dependence on the variable dimension only appears during the Kd-tree search but has a limited impact on the complexity. Concerning DGA, the impact of the variable dimension is also limited and only intervenes when computing the distance between the time steps. In the case of FFNN, however, the model dimension is directly related to the number of parameters to train and can have a significant impact on the performance of the network. For RC, the scaling with the model dimension is even worse since this dimension appears in the formula of the size of the reservoir, which is computed through a binomial product, so it may quickly blow-up.*

   We will provide a more detailed discussion of the impact of the model dimension on each method in the "Discussion" section, on page 26.

4. **The discussion about computational time is rather long for all models. I would suggest adding theoretical scalings of, say $O(N)$, $O(N^2)$, etc. to figure 3 and 5 to make a clear visualization and comparison.**

   *This is a very good suggestion. We call $N$ the number of samples in the training set. Due to the eigenvector computation, AMC has a complexity $O(N^2)$. For DGA, it is also because of an eigenvector problem that the scaling is $(O(N^3))$. The complexity of both FFNN and RC only depends on their architecture and they are given the training samples sequentially, hence a complexity in $O(N)$. If we now call $M$ the number of test samples, both FFNN and RC have a testing time scaling as $O(M)$ (they see the samples sequentially). Testing AMC mainly depends on building a Kd-tree from the training samples, hence its testing time scales at worst as $O(N \log N)$ and testing DGA requires extending the trained modes, which scales as $O(MN)$.*

   Adding these scalings to figures 3 and 5 may overload them with information. Instead, we will provide details about these scalings in pages 17-18 at the beginning of the discussion about computational times.